

# Low regeneration of lesions produced by coring in *Orbicella faveolata*

Rosa E. Rodríguez-Martínez[1], Adán Guillermo Jordán-Garza[1,2] and Eric Jordán-Dahlgren[1]

[1] Instituto de Ciencias del Mar y Limnología, Universidad Nacional Autónoma de México, Puerto Morelos, Quintana Roo, México

[2] Current affiliation: Universidad Veracruzana, Facultad de Ciencias Biológicas y Agropecuarias, Tuxpan, Veracruz, México

## ABSTRACT

The extraction of tissue-skeleton cores from coral colonies is a common procedure to study diverse aspects of their biology, water quality or to obtain environmental proxies. Coral species preferred for such studies in Caribbean reefs belong to the genera *Orbicella*. The long term effects of coring in the coral colony are seldom evaluated and in many Caribbean countries this practice is not regulated. We monitored 50 lesions produced on *Orbicella faveolata* colonies by the extraction of two centimeter-diameter cores to determine if they were able to heal after a four year period. At the end of the study 4% of the lesions underwent full regeneration, 52% underwent partial regeneration, 14% suffered additional tissue loss but remained surrounded by live tissue, and 30% merged with dead areas of the colonies. Given the low capacity of *Orbicella faveolata* to regenerate tissue-skeleton lesions, studies that use coring should be regulated and mitigation actions, such as using less destructive techniques and remediation measures after extraction, should be conducted to facilitate tissue regeneration.

## INTRODUCTION

The extraction of tissue-skeleton cores (cores) from reef-building coral colonies is a common procedure to study different aspects of their biology, such as growth rate (*Hudson, 1981*), calcification (*Carricart-Ganivet et al., 2012*), or the effect of diseases (*Closek et al., 2014*), or to study skeletal environmental proxies such as climate change (*Linsley et al., 2004*), paleo-nutrient proxies (*Mason et al., 2011*), water quality (*McCulloch et al., 2003*), or diagenesis (*Müller, Gagan & McCulloch, 2001*). One of the preferred coral species used for such studies in Caribbean reefs is *Orbicella faveolata*, due to its importance as a reef builder and because their colonies can attain relatively large sizes and thus record information from tens to hundreds of years (*Lough, 2010*). A literature search on Google Scholar (http://www.scholar.google.com), showed that for the period between 2005 and 2015 there were 80 published peer reviewed and grey literature articles with the name *Montastraea faveolata* or *Orbicella faveolata* in the title (not considering those focused on fossil colonies), and that 23% of the studies involved the extraction of cores.

Corresponding author
Rosa E. Rodríguez-Martínez,
rosaer@cmarl.unam.mx

Cores were extracted either with a hammer and a steel-core or with a pneumatic drill, their diameters ranged from 1.5 to 10 cm, and their depth varied from a few centimeters to over one meter, depending on the study goals. Only in 22% of these studies did the authors mention that the holes left by the cores were filled with an artificial substrate (i.e., concrete plugs or epoxy) to facilitate the regeneration and expansion of the coral tissue, and none of the studies report a follow-up to determine if the injuries healed.

The extraction of cores from corals can deleteriously affect the remaining colony, as the lesion can enlarge due to predation, competition with other sessile organisms (i.e., algae, sponges, or tunicates), or by the effect of boring organisms or pathogens (*Kramarsky-Winter & Loya, 2000*). The potentially negative effect of this methodology is important considering that *O. faveolata* is an endangered species (IUCN Red list category) as its populations have suffered severe declines in the last several decades due to the synergistic effects of temperature stress, diseases (*Edmunds & Elahi, 2007*), deterioration of environmental quality (*Harvell et al., 1999*; *Daszak, Cunningham & Hyatt, 2001*), and competitive interactions with algae, cyanobacteria, bio-eroding sponges and other competitors (*Titlyanov et al., 2005*; *Bruckner & Bruckner, 2006*). Recovery of these populations is compromised as species of the *Orbicella annularis* (complex) are known for having low larval recruitment rates, slow growth rates (∼6.3–11.2 mm of vertical growth per year; *Hudson, 1981*) and moderate regeneration capabilities (*Meesters, Pauchli & Bak, 1997*; *Cróquer, Villamizar & Noriega, 2002*).

In some countries (e.g., United States and Panama) coral coring is regulated and researchers are required to plug the holes to minimize the damage and maximize tissue and skeleton recovery. In others (e.g., Mexico, Colombia) plugging the holes after coring corals is not regulated or enforced. This lack of control allows that local and visiting researchers skip remediation techniques, which might be discarded as time consuming and unnecessary.

Here, we evaluate the fate of lesions produced by the extraction of tissue-skeleton cores for research purposes in colonies of the coral *O. faveolata* in a shallow Mexican Caribbean reef. We evaluated lesion size and depth immediately after coring and after a four year period, in apparently healthy and in yellow-band disease colonies, to determine to what extent *O. faveolata* colonies can regenerate from this type of injury.

## MATERIALS AND METHODS

Between September 2010 and February 2011, 50 cores were extracted (by another research group for a genomic study), from 16 *Orbicella faveolata* colonies, all larger than 50 cm in diameter, on Puerto Morelos reef, Mexico (20°52′N, 86°52′W); 12 colonies were located in the back-reef (5 m deep) and four colonies were located in the fore-reef (7 m deep). The number of lesions produced within a single coral colony ranged from two to seven, and the distance between them ranged from 0.1 cm to 30.4 cm (mean = 6.0 cm, SE = 5.7 cm). A map was made indicating the position of each colony that was sampled within the reef site. A photograph of each colony was taken to indicate the position of the tissue and skeleton extracted by each core. The cores were obtained using a 2 cm circular steel-core and a hammer. Occasionally, additional injury during the coring process occurred,

resulting in the loss of a larger portion of tissue and skeleton. The lesion produced by each core was immediately photographed with a digital camera and the depth of the hole produced was measured *in situ* with a Vernier caliper. After extraction, the core holes were not filled. Lesions were photographed again in May 2015. The software ImageJ was used to calculate the projected area of each lesion, using a 5-cm scale bar included in each image.

For the analysis, the cores were assigned to one of three sets: (1) 32 cores taken from 11 apparently healthy (H) colonies, (2) eight cores, taken from the yellow tissue of six colonies with yellow-band disease (YB), and (3) ten cores, taken from apparently healthy tissue on the same diseased colonies (hereafter called healthy-disease or HD). During coring, the healthy and the healthy-disease cores were always completely surrounded by live tissue, while the cores on the tissue with yellow band were not, due to disease-induced mortality adjacent to the yellow band. The percent change in the area of each lesion was estimated with respect to the original core measurement using the following formulas:

$$\Delta LA = LAt_0 - LAt_1 \tag{1}$$

$$\%\Delta LA = \frac{\Delta LA \times 100}{LAt_0} \tag{2}$$

with $\Delta LA$ = the change in lesion area ($cm^2$); $LAt_0$ = lesion area ($cm^2$) at time 0; $LAt_1$ = lesion area ($cm^2$) at time 1; and $\%\Delta LA$ = the change in lesion area expressed as a percentage from the original lesion. The sign of $\%\Delta LA$ indicates if tissue was lost ($\%\Delta LA < 0$) or if the lesion recovered ($\%\Delta LA > 0$).

Regeneration is expressed in terms of a reduction in lesion size. The possible outcomes of the lesions were: (a) full regeneration, (b) partial regeneration, (c) additional tissue loss, but remained completely surrounded by live tissue (Type I lesions: *Meesters, Pauchli & Bak, 1997*), (d) lesion enlarged and merged with a dead area of the colony (Type II lesions: *Meesters, Pauchli & Bak, 1997*) (Fig. 1).

The initial lesion areas were summed when they merged with adjacent core-produced lesions, with calculations treating each merged group as a single large lesion. When a lesion grew and merged with an area of the colony that lacked tissue it was excluded from the analysis because it was impossible to differentiate between the tissue loss associated to the core lesion and independent partial mortality.

Fieldwork was conducted within the Puerto Morelos Reef National Park under Permits DGOPA.10607.031009.3548 (in 2010), DGOPA.00322.200111.0099 (in 2011) and PPF/DGOPA-116/14 (in 2015), issued by the National Commission on Aquaculture and Fisheries (Comisión Nacional de Acuacultura y Pesca).

## RESULTS AND DISCUSSION

At the beginning of the study, the mean core–lesion area was 4.0 $cm^2$ (SD = 1.9), with a mean depth of 1.4 cm (SD = 0.8, range: 0.42–2.19 cm). After four years, two lesions (4%) underwent full tissue regeneration, 26 lesions (52%) underwent partial regeneration, seven lesions (14%) suffered additional tissue loss, but were still surrounded by live tissue,

## 2010-2011          2015

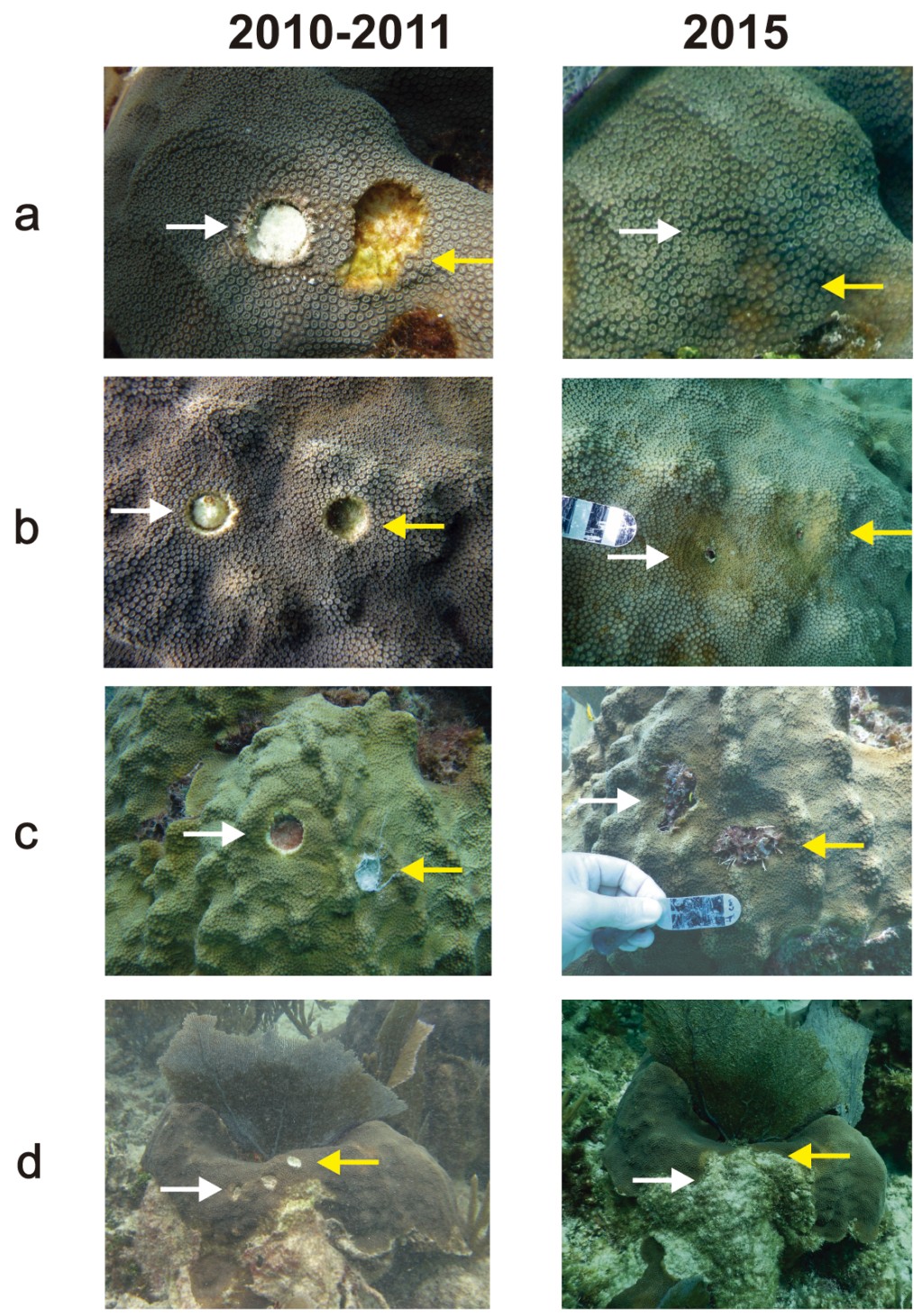

**Figure 1** Examples of different outcomes of tissue-skeleton core lesions in *Orbicella faveolata*: **(A)** full regeneration, **(B)** partial regeneration, **(C)** additional tissue loss, but still surrounded by live tissue and **(D)** lesion merged with a dead area of the colony and is no longer enclosed by live tissue. The photographs in the 2010–2011 column were taken between September 2010 and February 2011 and those in the 2015 column were taken in May 2015.

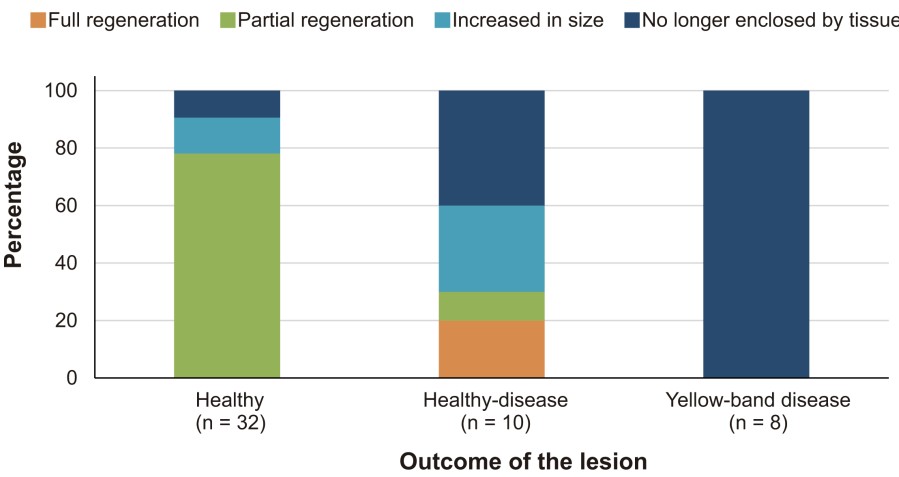

**Figure 2** Percentage of lesions that underwent full and partial regeneration of tissue and those that increased in size or were no longer enclosed by live tissue in healthy, healthy-disease, and yellow-band disease colonies between 2010–2011 and 2015.

and 15 lesions (30%) merged with a dead area of the colony and were no longer enclosed by living tissue.

After four years, none of the 32 lesions produced by the cores obtained from healthy-looking colonies underwent full regeneration, partial regeneration occurred in 78% of the cores which on average regenerated 61.9% (SD = 25.3%) of the original area produced by the lesion. The lesions produced by four cores increased in size (mean increment = 133.3 %, SD = 78.2%) and in three cases the lesions merged with a dead area of the colony and were no longer enclosed by living tissue in 2015 (Fig. 2). Of the ten lesions on apparently healthy tissue of colonies with yellow-band disease, two underwent full regeneration, another exhibited partial regeneration (regenerated area = 63% of the lesion), three fused and together increased in size by 579%, and four were no longer enclosed by living tissue in 2015 (Fig. 2). All the cores obtained from yellow-band diseased areas were no longer enclosed by tissue in 2015 (Fig. 2) due to the slow but persistent progress of this disease (*Bruckner & Bruckner, 2006*). The fact that the only two lesions that underwent full regeneration were those produced in the healthy tissue of one *O. faveolata* colony affected by yellow-band disease (Fig. 3) could be the result of chance, yet it is known that corals allocate their resources into three main hierarchical processes, growth, maintenance and reproduction (*Harrison & Wallace, 1990*) and when stressful situations occur (e.g., disease) tissue regeneration is frequently favored (*Henry & Hart, 2005*).

In all cases, the area of the coral colony that was cored appeared indented on the colony surface (Fig. 1B) suggesting that coral growth around the lesion was suppressed or hampered, as previously reported in *O. annularis* by *Meesters, Noordeloos & Bak (1994)*. The formation of septa, polyps, and internal skeletal structures likely results in reduced linear growth because the coral allocates resources to skeletal and tissue regeneration (*Henry & Hart, 2005*).

The identity of the coral colony had no apparent effect on the outcome of the regeneration of lesions, as lesions within the same colony showed a variable degree of regeneration

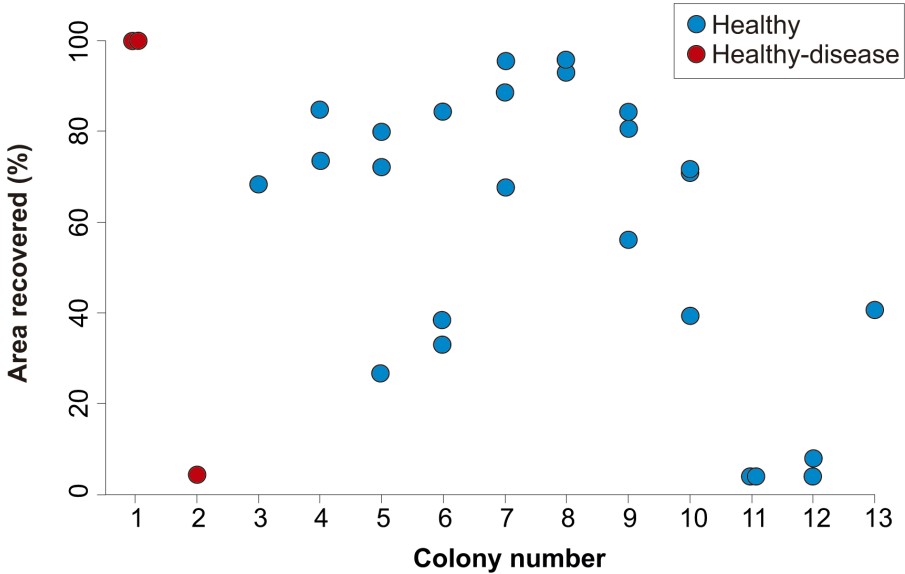

**Figure 3** **Percent area recovered four years after the extraction of tissue-skeleton cores from 13 colonies of *Orbicella faveolata*.** The number of lesions varied from one to three per coral colony. Only the colonies where lesion regeneration occurred are shown in the figure.

(Fig. 3). Although regenerative capacity has a genetic basis (*Meesters, Wesseling & Bak, 1996*), our observations suggest that, within the same colony, extrinsic factors (e.g., microhabitat, somatic mutations, different strains of zooxanthellae) can modulate the rate and success of lesion recovery.

The distances between lesions within a single colony didn't have an effect on the fusion of lesions; of the 64 cases in which colony fusion could occur, this happened in only nine cases and subsequent tissue loss occurred only in one case.

In 84.6% of the lesions that underwent partial regeneration, the coloration of the polyps surrounding the lesions was pale, suggesting a lower number of zooxanthellae or chlorophyll than in the rest of the colony (Fig. 1B). After observing a similar condition in *O. annularis* colonies weeks after the complete regeneration of artificially produced lesions, *Bak, Brouns & Heys (1977)* suggested that this was due to the expulsion of zooxanthellae. We propose that the pale coloration might also be associated with the presence of algae on unhealed lesions, especially when mixed turf algae (MTA) that trap sediments are present, as these have been reported to cause reductions in zooxanthellae densities and chlorophyll *a* concentrations in *O. faveolata* (*Quan-Young & Espinoza-Avalos, 2006*). In our study, MTA occupied unhealed lesions in 58% of the cases, calcareous coralline algae in 31% of the cases and fleshy algae in 11% of the cases. The tissue around the lesions was pale in 73.3% of unhealed lesions covered by MTA and in all lesions covered by calcareous coralline algae and fleshy algae. Further studies are needed to determine if the observed paling around lesions is indeed caused by the presence of MTA and other types of algae. Some benthic algae can also outcompete corals, increase coral stress, and are believed to act as reservoirs for a variety of different potential coral pathogens (*Sweet, Bythell & Nugues, 2013*) and contribute to additional tissue loss.

Given the low capacity of *O. faveolata* to regenerate lesions that involve the removal of tissue and skeleton (*Cróquer, Villamizar & Noriega, 2002*; *Sánchez et al., 2004*), we conclude that scientific studies that require the extraction of cores should design sampling protocols that minimize damage to colonies. Plugging core-holes with cement, epoxy or recycled skeleton from dead colonies in order to provide a hard substrate over which new coral tissue can spread may also prevent recruitment of boring organisms that can weaken the coral skeleton. This approach has allowed complete regeneration of tissue in some scleractinian coral species, such as *Pseudodiploria strigosa*, *P. clivosa*, and *Diploria labyrinthiformis* (*Weil & Vargas, 2010*), but not in others, such as *Meandrina meandrites* and *Montastraea cavernosa* (*Fahy et al., 2006*). In a study conducted by *Fisher et al. (2007)*, the filling of artificial lesions in *Orbicella* spp. with clay didn't prove to be effective, as only 13.1% of 229 lesions (area: 0.8–3.0 cm$^2$, depth: 3 mm) healed. These controversial results indicate that more studies are needed to find the best way to reduce long-term damage due to coring coral colonies. In the meantime, all countries with coral reef ecosystems should regulate this research technique and permits to employ it should establish mitigation actions to avoid damaging key coral species. Even if regulations are not established in a particular country researchers should use mitigation techniques whenever samples are obtained from this important and endangered species.

## CONCLUSIONS

*Orbicella faveolata* has low capacity to fully regenerate tissue-skeleton lesions produced by coring. Scientific studies that employ this sampling technique should minimize its effects by reducing the diameter and depth of cores and by plugging the holes. Environmental authorities from countries with coral reef ecosystems should regulate this sampling technique to reduce the impact from scientific studies on key reef-building species.

## ACKNOWLEDGEMENTS

This manuscript was greatly improved by comments from Carly Randall, Paul Blanchon, Bronwyn Rotgans and an anonymous reviewer.

### Funding

The authors received no funding for this work.

### Competing Interests

The authors declare there are no competing interests.

### Author Contributions

- Rosa E. Rodríguez-Martínez conceived and designed the experiments, performed the experiments, analyzed the data, wrote the paper, prepared figures and/or tables, reviewed drafts of the paper.

- Adán Guillermo Jordán-Garza analyzed the data, wrote the paper, reviewed drafts of the paper.
- Eric Jordán-Dahlgren conceived and designed the experiments, wrote the paper, reviewed drafts of the paper.

### Field Study Permissions

The following information was supplied relating to field studyfollowing approvals (i.e., approving body and any reference numbers):

Fieldwork was conducted within the Puerto Morelos Reef National Park under Permits DGOPA.10607.031009.3548 (in 2010), DGOPA.00322.200111.0099 (in 2011) and PPF/DGOPA-116/14 (in 2015), issued by the National Commission on Aquaculture and Fisheries (Comisión Nacional de Acuacultura y Pesca).

### Data Availability

Data can be found in the Supplemental Information.

### Supplemental Information

Supplemental information for this article can be found online at http://dx.doi.org/10.7717/peerj.1596#supplemental-information.

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
