# Peer review of "Low regeneration of lesions produced by coring in Orbicella faveolata"

_PeerJ, doi:10.7717/peerj.1596_

## Round 0.1 · original submission · Minor Revisions

the comments about experimental designs from both reviewer should be addressed.

Reviewer 1 ·

Basic reporting

Im not an expert on corals or marine biology. However, I find the study on long term consequence of injury and regeneration in Obricella by Rodriguez-Martinez is interesting and important for the field, and as reference for future international legislation concerning coring and similar procedures on corals. The authors find that coring injuries heal slowly in Orbicella and more worrisome is that the injuries may result in spreading of damage and even destruction of the colony.

Experimental design

Long-term study (four years) of regeneration after coring injuries to Orbicella (50 injury cores were followed over time). The Orbicella colonies were well documented and described. Overall the experimental design and statistical analysis is well described and makes sense.

Validity of the findings

Comments

Figure 2. Is not easy to follow and messy. What is pink and yellow bands in the bars? Would it be easier to represent all the conditions (full regeneration, partial reg….) in one bar (100%) as fractions for each of the conditions, for example, healthy. Its 100% stacked column in excel.

Any explanation why it seems like the healthy/disease corals regenerate better (full regeneration in Fig 2 and 3)?

Can authors elaborate more on the mechanisms that contribute to spreading of the damage? Infection

P 9. ….within same genets…Do you mean genetic background?

·

Basic reporting

Abstract lines 29-30: “such as using the less destructive possible technique…” the specific technique could simply be mentioned or something like “such as using less destructive techniques” should be used instead as it reads a little smoother.

Abstract lines 87-88: "to determine if this coral species can regenerate from this type of injury." There is a repetition of the same determiner, it would sound better to say either O. faveolata instead of "this coral species", or to say core-induced lesions instead of "this type of injury"

Materials and Methods lines 118-19: larger spaces ought to be incorporated between the equations and the "and" to make it easier to visually distinguish that they are two separate equations.

Results and discussion line 205: “…propose that the pale coloration also might also be associated with the…” redundant repetition of the word 'also', the first one should be deleted.

Results and discussion approx. lines 210-214: should probably mention either here or some other more appropriate place that algae are credited with inducing increased coral stress (potentially hampering lesion recovery) and that they are believed to act as reservoirs for coral diseases pathogens as per the 2013 Sweet et al paper "Algae as Reservoirs for Coral Pathogens"

In the Supplementary data: the contents of the column “lesion” are listed A-G, but referred to as cores in the “obs” column, (assuming this is referring to the same thing) these designations really should be consistent throughout the table.

In the Supplementary data: In the "obs" column, the entries referring to merged lesions should be included and repeated in each appropriate row eg “Fusion with cores A-C”.

In the Supplementary data: The acronym “WP” is included in the Obs column (rows 24-26) without being explained in the “codes” tab, I did not manage to find it explained anywhere in the manuscript as an acronym either.

Experimental design

Distances between lesions were not mentioned, they ought to have been recorded and included in the supplementary data. It seems entirely possible that a higher lesion density in a particular region could over-burden the corals regenerative capacity there and encourage tissue loss.

Materials and Methods lines 134-135 "When the lesions merged with adjacent core-produced lesions their areas were summed." This could be elaborated upon or re-stated more clearly, I initially read it as a self-evident observation on the process of lesions merging with one another. Something like the following could help to clarify the process: “The initial lesion areas were summed when they merged with adjacent core-produced lesions, with calculations treating each merged group as a single large lesion”

Validity of the findings

No Comments

Additional comments

The study itself is great in my opinion, I just picked up a few things going through it.

---

## Round 0.2 · accepted · Accept

While the revisions were acceptable, please double check the reference format before publication.

Reviewer 1 ·

Basic reporting

I have not further comments and the authors have done a good job in improving the manuscript and address issues.

Experimental design

NA

Validity of the findings

NA

Additional comments

NA

·

Basic reporting

No Comments

Experimental design

No Comments

Validity of the findings

No Comments

Additional comments

It's a very interesting area of research and I sincerely hope your paper contributes to changed practices in research communities who utilize coral coring in their studies.